# The Interpreter Understands Your Meaning:
# End-to-end Spoken Language Understanding Aided by Speech Translation

**Mutian He**[1,2], **Philip N. Garner**[1]
[1] Idiap Research Institute, Martigny, Switzerland
[2] Ecole Polytechnique Fédérale de Lausanne, Switzerland
{mutian.he,phil.garner}@idiap.ch

## Abstract

End-to-end spoken language understanding (SLU) remains elusive even with current large pretrained language models on text and speech, especially in multilingual cases. Machine translation has been established as a powerful pretraining objective on text as it enables the model to capture high-level semantics of the input utterance and associations between different languages, which is desired for speech models that work on lower-level acoustic frames. Motivated particularly by the task of cross-lingual SLU, we demonstrate that the task of speech translation (ST) is a good means of pretraining speech models for end-to-end SLU on both intra- and cross-lingual scenarios.

By introducing ST, our models reach higher performance over baselines on monolingual and multilingual intent classification as well as spoken question answering using SLURP, MINDS-14, and NMSQA benchmarks. To verify the effectiveness of our methods, we also create new benchmark datasets from both synthetic and real sources, for speech summarization and low-resource/zero-shot transfer from English to French or Spanish. We further show the value of preserving knowledge for the ST pretraining task for better downstream performance, possibly using Bayesian transfer regularizers.

## 1 Introduction

Modern artificial intelligence is characterized by large pretrained language models (PTLMs) with strong language capabilities to be adapted to various downstream tasks. The success of PTLMs rests on carefully-designed pretraining tasks to bestow the capability we expect on the model. Current PTLMs are mostly trained on self-supervised tasks, which started from masked language modelling (MLM) and next sentence prediction (NSP) in BERT (Devlin et al., 2019), but recently evolved into more difficult ones such as whole word (Cui et al., 2021) or span masking (Joshi et al., 2020),

text infilling, and token deletion (Lewis et al., 2020). While the rather simple NSP has been replaced by sentence permutation, document rotation (Lewis et al., 2020), and sentence order prediction (Lan et al., 2020). All those efforts introduced more challenges in the pretraining phase to mine stronger semantic supervision signals out of unlabelled data.

Such semantic-rich supervision is particularly relevant for pretrained spoken language models like wav2vec2 (Baevski et al., 2020) and HuBERT (Hsu et al., 2021) based on MLM on (sub-)phonetic units from lower-level audio signals, which are less informative and require models to carry out additional labor on acoustics. Therefore, their high-level capacities are more restricted. This may explain why automatic speech recognition (ASR) models finetuned upon them with paired data still have a role in fully end-to-end (E2E) SLU, often as a pretrained feature extractor (Seo et al., 2022; Arora et al., 2022). Unlike the cascaded SLU in which ASR produces transcripts for text processing, in such E2E systems ASR as an auxiliary or additional pretraining task provides strong supervision to explicitly link audio to representations that correspond to the denser and semantic-richer textual space, which is valuable for downstream understanding tasks.

On texts, self-supervised objectives are rather effective thanks to enormous text data with high information density, but supervised tasks are still used in many cases, machine translation (MT) being an often-seen one. A pioneer of the current PTLM paradigm, CoVe (McCann et al., 2017), is a seq2seq model pretrained on MT that achieved the then state-of-the-art on various downstream tasks. Belinkov et al. (2020) further validate language capabilities of MT on morphological, syntactic, and semantic levels, and T5 (Raffel et al., 2020) uses an ensemble of supervised tasks including MT. Furthermore, when trained with inputs of multiple languages, the model encoder may align and push representations for inputs in different lan-

guages with similar meaning together to have the same output in the target language, thanks to the guidance from paired data (Johnson et al., 2017; Schwenk and Douze, 2017). With this semantic-centric language agnosticity, such an encoder can achieve few/zero-shot transfer to another language in downstream tasks (Eriguchi et al., 2018).

Inspired by those works, we hypothesize that the counterpart of multilingual MT on speech, i.e., E2E multilingual speech translation (ST) that directly maps speech of various languages to texts in other languages, will also be effective as a pretraining task on E2E SLU, for three critical advantages:

1. It requires high-level understanding as an interpreter must "understand" the utterance before interpreting it into a different language, unlike ASR that transcribes speech verbatim and MLM on phonetic units that needs less semantic understanding.

2. It captures long-term dependency and a global view of the full input, in contrast to ASR and MLM which can often be resolved with local context.

3. It enables better cross-lingual transfer in comparison with multilingual ASR models and self-supervised PTLMs without the supervision that promotes language agnosticity.

Admittedly, ST data is only available in a limited number of language pairs, but for each covered language, there are infinite number of diverse downstream SLU tasks with only rich data in English. It is a practical need to enroll various such languages to an English-only model trained on each specific SLU task. Therefore, as shown in Figure 1, we may pretrain the model on speech translation between English and the target language French in both directions (i.e. En↔Fr), and then fine-tune on downstream tasks with an additional classifier, reusing the encoder. We show the benefit of our method on a variety of tasks for semantic understanding of speech, including mono- & multilingual intent classification (IC), spoken question answering (SQA), as well as speech summarization, for which we create a synthetic dataset following Huang et al. (2022). Then we show the strong advantage on cross-lingual transfer to French. All the experiments are focused on comparing ST with other tasks like ASR as the pretraining or auxiliary task to verify our core hypothesis above. In addition, to show that our method applies to other languages as well, we also conducted experiments using Spanish as the target language. This is evaluated by creating the French and Spanish version of the English IC

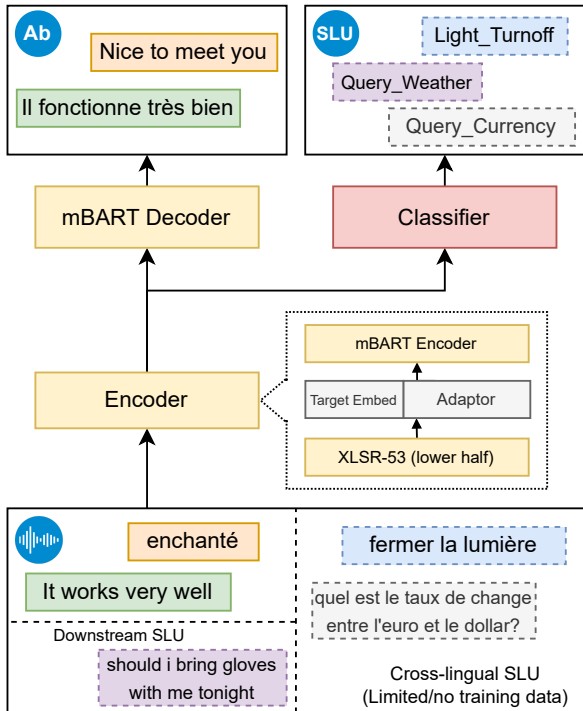

Figure 1: Our framework of ST-aided SLU, by connecting pretrained XLSR and mBART fine-tuned for ST following Li et al. (2021), and then reusing the ST encoder, transferred to downstream SLU tasks like intent classification with a stacked classifier also from PTLMs.

benchmark SLURP (Bastianelli et al., 2020), using both real and synthetic sources. On all the tasks, our approach outperforms previous baselines and ASR pretraining, often by a large margin.

Furthermore, unlike knowledge for self-supervised objectives loosely connected to target SLU tasks, knowledge to handle tasks with closer link to semantics such as ST will be more valuable, following our core hypothesis. Hence it should be helpful to preserve such knowledge instead of direct fine-tuning with the risk of catastrophic forgetting. Therefore, we introduce multi-task learning as well as Bayesian regularizers for knowledge preservation, namely L2-SP (Li et al., 2018b) and EWC (Kirkpatrick et al., 2017), which show benefits especially in low-resource cases.

To summarize, our contributions are three-fold:

1. We demonstrate the effectiveness of speech translation pretraining on multiple SLU tasks, especially in cross-lingual transfer cases.

2. We confirm the value of preserving ST pretraining knowledge for downstream tasks and the capability of Bayesian regularizers to achieve that.

3. We build several new datasets for speech summarization and cross-lingual SLU.

Our code, models, and datasets will be released at .

## 2 Model Pretraining

As in Figure 1, we first build a speech translator using an architecture established by Li et al. (2021), that connects pretrained models on speech and text with a CNN adaptor: Audio signals are fed into the lower half of the multilingual wav2vec2, XLSR-53 (Conneau et al., 2021), to extract the (sub-)phonetic representations into a 320x-downsampled sequence. The upper half (12 layers) of XLSR is discarded for computational efficiency, as those parameters are found focused on MLM pretraining and less useful for downstream tasks (Zhu et al., 2022). Given the phonetic level embeddings produced by the half XLSR, the task is similar to machine translation to map them to the output text, for which we leverage the MT model based on mBART (Liu et al., 2020). While the length of this sequence is still much longer than the corresponding text. To better align it with typical textual embeddings as in mBART inputs, a 3-layer 8x-downsampling CNN adaptor is inserted. A *target embedding* is then prepended to specify the target language or task, similar to the target token used in mBART. To promote language agnosticity, we do not indicate the source language. Furthermore, it has been found that explicitly promoting language agnosticity may help zero-shot transfer (Arivazhagan et al., 2019), hence we've also attempted to add language adversarial training on the encoder outputs during pretraining and fine-tuning, using a language classifier of a 2-layer MLP to predict the language of the input speech, with gradient reversal layer to explicitly align the representations between different languages.

Based on the architecture, we fine-tune the model using a combination of the En→Fr portion of MuST-C (Gangi et al., 2019), and the Fr→En portion of TEDx (Salesky et al., 2021), both derived from TED talks, plus the Fr→En portion of CoVoST2 (Wang et al., 2021a) based on general sentences in Common Voice (Ardila et al., 2020), with texts further cleaned and sentences that are too long or contain foreign characters removed. Unlike Li et al. (2021), the whole model is fine-tuned for best pretraining results. To compare, we also experiment with pretraining on the task of ASR instead. As the data are paired with both translations and

| ASR WER↓ | TEDx | MuST-C | CoVoST2 |
|---|---|---|---|
| ASR | 16.58% | 8.62% | 13.67% |
| ASR+ST | 15.82% | 8.28% | 13.62% |
| **ST BLEU↑** | | | |
| ST | 29.27% | 36.30% | 31.34% |
| ASR+ST | 31.19% | 37.18% | 31.93% |

Table 1: Test results on the cleaned pretraining datasets given by word error rate (WER) for ASR and BLEU score for ST, with French inputs for TEDx and CoVoST2, and English inputs for MuST-C.

transcripts, we use the same ST dataset for ASR training to build a multilingual (En+Fr) ASR model. We've tried to jointly train on ASR+ST in a multi-task manner as well. With a total of >700 hours paired speech data, we achieve satisfactory results on the pretraining tasks as indicated in Table 1, and ASR+ST training shows better performance compared to the single-task ones. Starting from the ASR and ST models, we further add Spanish portion from the same set of ST datasets. As a result, we obtain an ST model supporting both En↔Fr and En↔Es (Spanish), and a tri-lingual En+Fr+Es ASR model, both with similar satisfactory results, details available in Appendix E.

## 3 Downstream Adaptation

### 3.1 Tasks

We then fine-tune the whole model on a variety of direct downstream tasks as follows.

**SLURP** is a large and challenging English SLU dataset recently proposed, with 72.2k real speech recordings and 69.3k synthetic audio for a broad range of speech commands given to voice assistants. We use its IC labels to classify the input into 18 scenarios and 46 actions.

**MINDS-14** is a multilingual IC dataset for banking scenarios with 14 types of intents in 14 languages with around 600 utterances per language, and we use four subsets (en-AU, en-GB, en-US, and fr-FR) under a 3:2:5 train-dev-test split in XTREME-S (Conneau et al., 2022). The rather scarce training data demand data-efficient multilingual modelling.

**NMSQA** or Natural Multi-Speaker Question Answering is a spoken QA dataset consisting of audio for the questions and segmented context articles from SQuAD (Rajpurkar et al., 2016), with 97.6k

question-answer pairs given in >300 hours of synthetic audio from 12 speakers produced by Amazon TTS, coupled with a 60-speaker real test set of 2.7 hours of recordings. In this task, the goal is similar to textual QA to predict the correct span in the spoken context audio that answers the question, and the performance is measured by Audio Overlapping Score (AOS) (Li et al., 2018a), defined as $AOS = X \cap Y / X \cup Y$, in which $X$ is the predicted audio span and $Y$ the ground truth.

**Spoken Gigaword** is the synthetic spoken version of the summarization or headline generation task on Gigaword (Rush et al., 2015), proposed by Huang et al. (2022), aimed at generating a brief headline from a short piece of English spoken news. As it was not released, we follow their method to filter the data and create a synthetic dataset of 131.5 hours of audio using Google TTS from 9 neural voices in en-US, with 50k training samples, 1k validation samples, and 385 test samples, as a result of filtering out the frequent noise in the test set.

Synthetic data are used in those established datasets for training and evaluation. Despite being possibly different from real data, it is observed that they are often reliable to reflect model performances and well correlated with real cases.

## 3.2 Methods

For these downstream tasks, we reuse the encoder further pretrained on ST/ASR (with French, unless otherwise stated). It should be noted that the 12-layer mBART encoder we use is slightly smaller than half XLSR. So when connected, the total encoder size and the computational cost to fine-tune it is comparable with fine-tuning the whole original XLSR. Upon the encoder we stack a 3-layer transformer, which is also transferred from a PTLM. As for IC, we use layer 2-4 from pretrained XLM-R (Conneau et al., 2020) for possibly better understanding capabilities, stacked with linear classifier heads over mean-pooled outputs. Particularly, for SLURP in which the intent consists of a scenario and an action, two heads are used. As for SQA, we use layer 2-4 of pretrained Longformer (Beltagy et al., 2020), a PTLM dedicated for long utterances due to the length of each segment in the data, as in Lin et al. (2022). Two linear classifiers are then applied to each frame to predict the start and end of the span, along with an answer existence classifier over mean-pooled outputs to predict if the answer exists in the provided segment. We then concatenate the question audio with each segment in the spoken article as model inputs, and pick the predicted answer span from the segment with the highest answer existence likelihood. For these two tasks, the pretrained decoder is simply discarded. While speech summarization is more similar to ASR and distinct from other downstream tasks that the model first needs to capture general meaning of the speech as encoded representations, and then generate a textual summary by the decoder, which demands a seq2seq architecture identical to the ST/ASR pretraining task. Hence we reuse the whole encoder-decoder model and formulate the task as generation in an extra "target language". With the needs to both understand the general meaning and generate in the same language, we hypothesize that combining ASR and ST will lead to the best results.

Furthermore, as mentioned above, direct model fine-tuning may lead to catastrophic forgetting of the knowledge on ASR or ST and harm semantic understanding capabilities. Hence we also tried a multi-task *joint training* approach on both the pretraining and target task. Results are compared between the model pretrained with ST, ASR, or both, or one directly derived from self-supervised pretraining without further supervision (***None***), plus other baselines. More, the recent Whisper (Radford et al., 2023) is trained on multiple speech tasks including ASR and ST, which matches our idea despite not aiming at SLU. Hence we also try to fine-tune the Whisper encoder, using the *medium* version with size similar to our encoder.

## 3.3 Results

**English IC** Following the previous works, we report the test accuracy on SLURP as in Table 2. It can be observed that the models with ST pretraining outperform those trained on ASR only, while adding ASR to ST pretraining makes limited improvements, though it gives better WER and BLEU during pretraining; it is the same case for the model with the extra Spanish ST task introduced in pretraining. However, ASR does help considering the *None* model directly fine-tuned from self-supervised PTLMs without any additional pretraining. By joint training with both the pretraining and downstream task, results are consistently improved. However, despite being a strong ASR+ST model, Whisper is found not suitable for fine-tuning on SLURP in this way as shown by the low accuracy.

| Pretraining task | Accuracy↑ |
|---|---|
| ASR | 87.38% |
| w/ Joint training | 88.37% |
| ST | 87.84% |
| w/ Joint training | 89.35% |
| w/ Joint training + Es | **89.59%** |
| ST+ASR | 87.75% |
| w/ Joint training | 89.43% |
| None | 84.80% |
| Whisper | 80.39% |
| ESPnet-SLU (Arora et al., 2022) | 86.30% |
| CTI (Seo et al., 2022) | 86.92% |
| Generative IC+SF (Wang et al., 2021b) | |
| based on wav2vec2 | 87.13% |
| based on HuBERT | 89.38% |
| CIF-PT (Dong et al., 2023) | 91.43% |

Table 2: SLURP test results of our models, fine-tuned from wav2vec2, compared to baselines without additional supervised pretraining or reusing Whisper as the encoder, as well as results reported in literatures.

This might be explained by the fact that Whisper is trained on En ASR and X→En ST but not for En→X ST. Our hypothesis is that the ST pretraining on a specific language would enhance the semantic understanding capabilities of the model in that language, which may not help Whisper much on the English SLURP benchmark. Also, Whisper is trained on 30-second chunks, while SLURP contains more shorter utterances.

HuBERT, used in multiple baselines in Table 2, has been found stronger on various downstream tasks compared to wav2vec2. Owing to the lack of a multilingual HuBERT (large) model, we rely on the multilingual wav2vec2 as our acoustic encoder. However, we reach much better results compared to many notable baselines, including the approach of jointly generating the intents and slots (Wang et al., 2021b), with 87.13% accuracy, the highest among wav2vec2-based baselines. We also reach slightly higher accuracy than its HuBERT version, which was the previous state-of-the-art. The very recent CIF-PT (Dong et al., 2023), concurrent with ours also injects more semantic signal, but by learning frame-to-token alignments on the encoder and then distilling from PTLMs, significantly pushing the state-of-the-art on this monolingual benchmark. Nevertheless the method is distinct from ours, raising the possibility of applying both methods or-

thogonally for further improvement, and we maintain advantages on cross-lingual transfer and possibly also generative tasks by reusing a pretrained seq2seq decoder, as elaborated below.

**Multilingual IC** We then report the accuracy on MINDS-14 as in Table 3 on four languages plus the average accuracy across languages, compared to a baseline directly fine-tuned from XLSR. The results are consistent with the monolingual case that ST pretraining can significantly improve the performance on SLU tasks, that joint training is beneficial, and that adding ASR gives limited gains.

**Spoken QA** We compare our methods with results reported by Lin et al. (2022), including the results from a cascaded pipeline that fine-tunes Longformer upon transcripts from wav2vec2-based ASR, and the DUAL approach that fine-tunes Longformer upon units pre-extracted by a frozen HuBERT, hence not fully end-to-end. For fair comparison, we fine-tune the classifier built by layers 2–4 of Longformer and the top 5 layers of the mBART encoder, while the rest of the model is frozen and used as a feature extractor, so that they have a comparable number of trainable parameters with the baselines. Therefore we do not conduct experiments on joint training in this task as most shared parameters are frozen. The results reported in the more recent T5lephone (Hsu et al., 2023) including the E2E and cascaded approach are also mentioned, though they are almost twice as large as other models. All the baselines enjoy a view of the whole article, while in our experiments we use a model that works on a shorter context window with the question and each segment in the article individually, in order to have an end-to-end architecture given our computational resources that is consistent with other experiments. Therefore, the baselines possess a strong advantage over ours. However, as shown in Table 4, the additional pretraining stage leads to better results compared to all the E2E baselines, which further demonstrates the advantage of our approach. Particularly, ST considerably improves the performance and could successfully beat the cascaded system reported by Lin et al. (2022) in the more challenging *test* portion.

**Speech summarization** We report the results compared between different auxiliary tasks as in Table 5 using the ROUGE-1/2/L metrics (Lin, 2004). In the experiments, we observed that simply fine-tuning the model rapidly leads to overfitting, hence

| Pretraining task | en-AU | en-GB | en-US | fr-FR | Average |
|---|---|---|---|---|---|
| ASR | 95.7% | 97.3% | 96.5% | 95.2% | 96.2% |
| w/ Joint training | 96.3% | 98.3% | 98.2% | 93.7% | 96.6% |
| ST | 96.9% | **99.0%** | 98.2% | 97.8% | 98.0% |
| w/ Joint training | **97.3%** | 98.7% | **99.3%** | 98.2% | **98.3%** |
| ST+ASR | 95.4% | 98.3% | 97.5% | 95.6% | 96.7% |
| w/ Joint training | 96.3% | 98.3% | 98.9% | **98.5%** | 98.0% |
| XLSR (Lozhkov, 2022) | 92.4% | 93.2% | 93.3% | 94.4% | 93.3% |

Table 3: Test accuracies for models on MINDS-14 multilingual IC, comparing with directly fine-tuning the full XLSR model. Both ST pretraining and joint training show benefits.

| Pretraining task | *dev* | *test* |
|---|---|---|
| ASR | 54.6% | 53.0% |
| ST | **58.2%** | **59.4%** |
| ST+ASR | 57.8% | 58.0% |
| DUAL E2E | 48.5% | 49.1% |
| - Cascaded | 58.3% | 57.4% |
| ByT5lephone E2E | - | 53.3% |
| - Cascaded | 59.2% | 70.5% |

Table 4: AOS (↑) scores for models on NMSQA, compared to baselines reported in DUAL (Lin et al., 2022) as well as the much larger ByT5lephone model (Hsu et al., 2023). The pretraining tasks prove helpful, particularly ST pretraining, which may reach performance close or better certain cascaded system.

we perform joint-training only, and use a special target embedding to indicate the summarization task. ASR is helpful on the summarization task as the ST+ASR model consistently outperforms the ST one, while the ST one is still better than the ASR-only model, signifying the importance of the semantic understanding capability brought by ST pretraining. In addition, we compare with a cascaded baseline that first transcribes the inputs with our ASR model, which introduces WER of 9.1% and 8.9% on *dev* and *test* respectively. Then we leverage a BART-based model fine-tuned on the full textual Gigaword with ROUGE-1/2/L=37.28/18.58/34.53 to produce the summaries. When applied to the relatively simple utterances in Spoken Gigaword, it reaches a higher performance on *dev*, which suggests the challenges for E2E systems in our benchmark, though the gap is narrow compared to our E2E approach with ST+ASR pretraining, and on the noisier *test* set our E2E models consistently get much better results.

## 4 Cross-lingual Transfer

For cross-lingual transfer, IC models trained on SLURP are then applied on/fine-tuned to French/Spanish data below:

**Datasets** A French version of SLURP, **SLURP-Fr**, is created to evaluate the cross-lingual transfer capabilities of the model, which is based on MASSIVE (FitzGerald et al., 2023), a translation of SLURP texts into multiple languages. With the same input domain and output categories, zero-shot transfer becomes possible. We first produce the audio for the 16.5k French samples in MASSIVE with a 7:2:1 train-dev-test split using Google TTS from four different WaveNet-based speakers. Then we invite two native French speakers to read out a total of 477 randomly-selected category-balanced held-out utterances, forming the *real* test set. To mimic SLURP, we record the audio indoors with two microphones under both near-field and far-field conditions. We also define a 100-shot per category subset with 4.5k samples in total to simulate a condition with even lower resource. **SLURP-Es** is created in a way similar to SLURP-Fr with 16.5k Spanish samples in MASSIVE, though we are unable to create a *real* set.

**Experiments** The advantage of our method on cross-lingual transfer is evaluated under the full-data, 100-shot, and zero-shot cases, using different pretraining strategies compared to the *None* model trained on SLURP without further supervision but directly upon the multilingual self-supervised pretrained models. Hence it is noteworthy that all the compared models have been pretrained in a multilingual way. As given in Table 6 on French, extra multilingual ST/ASR supervision consistently leads to better results on different data amounts. ST pretraining outperforms ASR, similar to previ-

| Joint task | dev | | | test | | |
|---|---|---|---|---|---|---|
| | ROUGE-1 | ROUGE-2 | ROUGE-L | ROUGE-1 | ROUGE-2 | ROUGE-L |
| ASR | 40.16 | 18.39 | 37.69 | 35.90 | 16.25 | 33.77 |
| ST | 40.61 | 18.95 | 38.23 | 36.70 | 16.39 | 34.47 |
| ST+ASR | **41.39** | **19.50** | **38.83** | **37.63** | **17.80** | **35.20** |
| None | 21.49 | 7.44 | 20.37 | 18.39 | 6.16 | 17.41 |
| Cascaded | **42.00** | **21.42** | **39.60** | 32.24 | 15.03 | 30.14 |

Table 5: ROUGE (↑) scores for models on Spoken Gigaword speech summarization. ST still proves beneficial, while best results could be obtained by combining ASR for this task of generating summaries in the same language.

| Pretrain task | Full | | | 100-shot | | | Zero-shot | | |
|---|---|---|---|---|---|---|---|---|---|
| | dev | test | real | dev | test | real | dev | test | real |
| ASR | 83.3% | 84.0% | 79.7% | 69.6% | 69.0% | 71.3% | 39.0% | 39.8% | 39.4% |
| ST | 85.2% | **86.1%** | **84.9%** | 78.1% | 77.0% | 79.0% | 58.9% | 58.9% | 56.6% |
| ST+ASR | 85.8% | 85.7% | 82.4% | 78.0% | 77.0% | 78.8% | 63.9% | 62.6% | 59.1% |
| ST+Adv. | **86.4%** | 84.9% | 84.1% | **78.3%** | **78.1%** | **80.9%** | **67.0%** | **67.7%** | **63.7%** |
| None | 75.9% | 74.0% | 65.4% | 57.5% | 52.4% | 53.5% | 15.3% | 16.1% | 13.6% |

Table 6: Results on SLURP-Fr cross-lingual IC transferred from SLURP with different data amounts. Comparing different supervised pretraining (or w/o additional supervision), results highlight ST and adversarial training.

| | | None | ASR | ST |
|---|---|---|---|---|
| Full | dev | 75.6% | 83.7% | 85.6% |
| | test | 75.1% | 82.5% | 84.5% |
| 100-shot | dev | 64.4% | 76.5% | 80.3% |
| | test | 63.8% | 75.1% | 79.6% |
| Zero-shot | dev | 12.3% | 39.7% | 55.2% |
| | test | 12.8% | 39.1% | 54.4% |

Table 7: Results on SLURP-Es cross-lingual IC transferred from SLURP with different data amounts.

ous experiments, while ST+ASR joint pretraining brings some improvements in the zero-shot case. Notably, the gap between ASR and ST models becomes larger with fewer data, especially with zero shot, which implies the importance of ST on cross-lingual transfer. Accuracy on the real near-field speech is reported, which correlates well with those on synthetic ones, indicating that performance on synthetic speech is reliable for evaluation. The *ST+Adv.* model incorporates language adversarial training during pretraining and fine-tuning to further promote language agnosticity as mentioned above, which outperforms other models in most cases, particularly with zero shot, implying the usefulness of language adversarial training and the importance of language agnosticity of input

features to the classifier. In addition, we build a cascaded system which first translates the speech into English text using our ST model, with BLEU scores of 26.08/26.34/21.56 on dev/test/real. Then a BART-based textual SLURP model with 85.7% test accuracy is used. This essentially zero-shot system gives a competitive 62.2% *real* accuracy, which poses challenges to the future development of E2E cross-lingual models. The model on Spanish, based on En+Fr+Es ST/ASR pretraining along with training on SLURP in an identical protocol, shows similar results as in Table 7, indicating the applicability of our approach to other languages.

## 5 Pretraining Knowledge Preservation

**Methods** As mentioned above, the knowledge to perform ASR/ST that connects speech and semantic-rich texts could be valuable for downstream tasks, which motivates us to use *joint training* above that maintains the performance on the pretraining tasks. This is verified by the considerable performance improvement or gap between the joint and single models. However, it is computationally intensive and requires access to the pretraining data. To alleviate the gap without joint training, we intend to explore Bayesian transfer/continual learning regularizers that limit the parameter shift by applying a prior on the parameters,

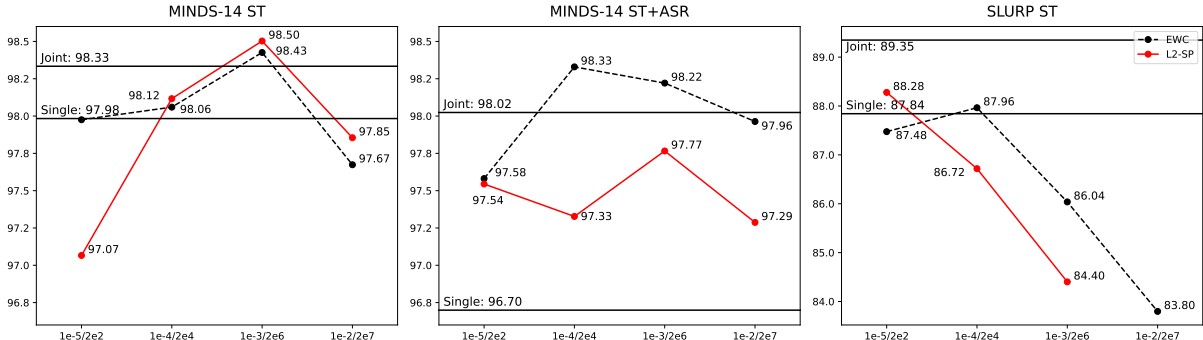

Figure 2: Results for Bayesian transfer regularizers when applied to different tasks, with the goal of mitigating the gap between the performance of single-task and joint-training models, indicated by the lower and the upper horizontal lines. The x-axis indicates the regularization weight for EWC/L2-SP, and the y-axis the accuracy. The regularizers bring positive effects on the data-scarce MINDS-14 task, but not on SLURP.

based on the Laplacian approximation of posterior parameter distributions in pretraining (MacKay, 1992). Particularly, the L2-SP method formulate the prior as an isotropic Gaussian distribution with the pretrained parameters $\theta_0$ as the mean and identical variance for all parameters, which leads to an L2 regularization term with weight $\alpha$ centered at $\theta_0$ in the loss for the maximum likelihood estimation of the parameters (Li et al., 2018b). While elastic weight consolidation (EWC) (Kirkpatrick et al., 2017) considers the variance of each parameter $\theta_i$ decided by the Fisher diagonal $F_i$, which could be further estimated using squared gradients by averaging over the stochastic gradient descent (SGD) trajectory. However, for optimization we use the Adam algorithm

$$\theta_t \leftarrow \theta_{t-1} - \alpha \cdot \hat{m}_t / (\sqrt{\hat{v}_t} + \epsilon), \qquad (1)$$

that already computes $\hat{v}_t$, an exponential moving average of squared gradients (Kingma and Ba, 2015), close to linear averaging with a smoothing parameter $\beta_2 = 0.999$. Hence we reuse them to set the per-parameter weight $\alpha F_i$ for regularization. For both methods, the hyperparameter $\alpha$ is used to control the strength of the knowledge preservation or the restraint to the parameter update. See Appendix A for more theoretical explanations.

**Experiments** We experiment with these regularizers, targeted on ST-pretraining on SLURP and MINDS-14, plus the ST+ASR pretraining on MINDS-14 which has a considerable 1.32% accuracy gap. We try to use various weights $\alpha$ for L2-SP regularization ranging from 1e-5 to 1e-2. Then we inspect the distribution of the approximated $F_i$, which ranges from 1e-20 to 1e-5 as in Appendix F. For optimization stability we clamp

the weight $\alpha F_i$ above 1e-2, and use EWC weights of 2e2, 2e4, 2e6, and 2e7 to roughly match the magnitude of those for the L2-SP regularizer.

Results are shown in Figure 2, and for MINDS-14 the average accuracies are reported. In the case of SLURP, it is possible that the amount of data is already sufficient that the preservation of the pretraining knowledge could be helpful only if it is carried out in a fully adaptive way, namely joint training. Therefore, the regularizers lead to limited help or even harm to the accuracy when the weight is large. However, under the low-resource condition in MINDS-14, both regularizers are effective. As in Li et al. (2018b), although being more flexible and adaptive, EWC doesn't necessarily lead to better transfer learning. This is consistent with our observations: Both regularizers can successfully overcome the accuracy gap or even go beyond the joint training model under an appropriate weight, while the best regularizer varies in different cases, though the more adaptive EWC has a chance to reach better results as in the MINDS-14 ST+ASR case. In this way, we demonstrate the effectiveness of Bayesian parameter-preserving regularizers for transfer learning on such large pretrained models.

## 6 Related Work

**Translation as an auxiliary task** It has been found that representations from MT models capture various aspects of the input utterance such as syntax (Shi et al., 2016), morphology (Belinkov et al., 2017), and also semantic inferences (Poliak et al., 2018; Belinkov et al., 2020). Hence MT has been established as a pretraining task as in CoVe (McCann et al., 2017) for various downstream tasks. But unlike this paper, recent works

on the direction has been focused on multilingual and cross-lingual cases, starting from attempts to reuse MT representations as sentence embeddings for text classification (Shi et al., 2016; Lu et al., 2018), and, particularly often, for semantic similarity and bi-text mining (Schwenk and Douze, 2017; Vázquez et al., 2019; Raganato et al., 2019; Artetxe and Schwenk, 2019). As for pretraining PTLMs to be fine-tuned, MT proves effective for downstream cross-lingual tasks on few-shot and zero-shot transfer (Eriguchi et al., 2018), while often accompanied with similar tasks like translation language modelling (Conneau and Lample, 2019; Kale et al., 2021), cross-lingual MLM (Chi et al., 2021), and dictionary denoising (Reid and Artetxe, 2022). Particularly, MT has been used as an auxiliary task for cross-lingual intent classification on texts (Schuster et al., 2019; Siddhant et al., 2020; van der Goot et al., 2021), and is widely used on cross-lingual generation, including summarization (Zhu et al., 2019; Cao et al., 2020; Xu et al., 2020; Takase and Okazaki, 2022), simplification (Mallinson et al., 2020), question generation (Chi et al., 2020), and data-to-text generation (Kale and Roy, 2020).

**End-to-end SLU**  Cascaded SLU methods work on ASR transcripts, for which error propagation is a major challenge (Chang and Chen, 2022; Cheng et al., 2023a). Hence recently end-to-end methods have gained popularity (Serdyuk et al., 2018; Haghani et al., 2018), especially with the performance gap compared with cascaded systems mitigated in many cases thanks to the PTLM paradigm. Besides directly fine-tuning existing PTLMs on speech (Wang et al., 2021b; Arora et al., 2022), there are also explorations for end-to-end interface to connect pretrained models on speech and text (Saxon et al., 2021; Seo et al., 2022; Raju et al., 2022), as well as joint speech-text modelling, pre-training, or distillation (Chuang et al., 2020; Chung et al., 2021; Kim et al., 2021; Villatoro-Tello et al., 2023; Dong et al., 2023), prompt tuning for PTLMs (Gao et al., 2022; Chang et al., 2022), combining PTLM features (Cheng et al., 2023b), and multitask learning with ASR (Huang et al., 2022).

**Bayesian transfer learning**  Viewing the pre-trained model not as a point estimation but a distribution is critical for continual learning as in EWC (Kirkpatrick et al., 2017), and the idea has been also applied to transfer learning to regularize fine-tuning as in L2-SP for image classification (Li et al., 2018b), though similar regularizers have been used on MT (Barone et al., 2017) and ASR (Liao, 2013). More recently, Shwartz-Ziv et al. (2022) propose to approximate the prior using SGD trajectory as in SWAG (Maddox et al., 2019) for transfer learning.

## 7   Conclusion

We confirm our hypothesis that speech translation can be a powerful pretraining and joint-training means for various end-to-end models on tasks involving semantic understanding of speech. Particularly, it benefits multilingual scenarios and cross-lingual transfer, including the zero-shot case. We also create two new datasets for the above tasks. Furthermore, we demonstrate the effectiveness of Bayesian regularizers to preserve the knowledge from pretraining for downstream tasks.

## Limitations

Some of the limitations of our paper are:

1. The best results are mostly achieved with multi-task learning, which adaptively preserves the knowledge from the pretraining task, but much slower, computationally intensive, and energy consuming. Therefore we explore the regularizers from continual learning for knowledge preservation, while there are some other continual learning approaches (e.g. Learning without Forgetting, Gradient Episodic Memory) that might be helpful. Also, we haven't explored alternative regularization approaches and light-weight tuning.

2. On the monolingual case (i.e. on SLURP), despite getting much better result under fair comparison with the alternative training methods and other baselines based on wav2vec2, our result is only slightly better than the HuBERT-based generative approach (Wang et al., 2021b), which is the state-of-the-art before us. Very recently, CIF-PT (Dong et al., 2023), parallel with our work in time, reaches 1.9% higher than both types of models, marking a new state-of-the-arts. This approach appears to be orthogonal to ours and the two methods might be jointly applied to the SLU model to reach even better results, but this is left for future work.

3. The dataset we built is relatively small and with limited number of real samples.

## Ethics Statement

We honor the ACL Code of Ethics. Particularly, as our work involves data collection, we go through a

formal process at the institution for collecting audio data, strictly follow the general and local rules for data protection, and receive full consent of participants to process and release the data. Since cross-lingual transfer is highlighted in our work, the work could have positive societal impacts for the application of speech and language technology in the non-English population. We believe that there is little chance for the method to be misused, except in cases of misusing SLU, such as mass surveillance. We also emphasize the reproducibility, and will release relevant code and models.

## Acknowledgements

This work received funding under project SteADI, Swiss National Science Foundation grant 197479.

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

## A  Bayesian Transfer Learning

As in the standard machine learning configuration, we determine the parameters by optimizing loss with an L2 regularizer, i.e. minimizing $\mathcal{L}(D;\theta) + \alpha\|\theta\|_2^2$ for the parameters $\theta \in \mathbf{R}^N$ given data $D = \{(x,y)\}$ and the hyperparameter $\alpha$, in which the cross-entropy loss $\mathcal{L}$ corresponds to the negative log-likelihood $-\log p(y|\theta)$ of the label upon model outputs. This can be formulated as maximum likelihood estimation (MLE) of $\theta$ by maximizing $\log p(\theta|D)$, which is equal to $\log p(D|\theta) + \log p(\theta) - \log p(D)$ by Bayes' theorem. With constant $p(D)$ and a zero-mean isotropic Gaussian prior $\mathcal{N}(0, \sigma^2 I)$ on $\theta$ with scalar $\sigma$, the optimization objective corresponds to

$$
\begin{aligned}
\log p(\theta|D) &\propto \log p(D|\theta) + \log p(\theta) \\
&= \log p(D|\theta) + \log(\mathcal{N}(\theta; 0, \sigma^2 I)) \\
&\propto -\mathcal{L}(D;\theta) - \frac{1}{2\sigma^2}\sum_{i=1}^{N}\theta_i^2
\end{aligned}
\tag{2}
$$

Hence L2 regularization can be viewed as giving an isotropic zero-mean Gaussian prior to the model parameters that assigns higher probability to close-to-zero parameters, with a larger $\alpha$ indicating a smaller scalar $\sigma^2$. While instead of zero, L2-SP (Li et al., 2018b) proposes to limit the parameter shift from the pretrained ones during fine-tuning by assigning a Gaussian prior $\mathcal{N}(\theta_0, \sigma^2)$ centered at pretrained parameters $\theta_0$, which has been found to lead to better downstream performance.

Nevertheless, it is an over-simplification of the prior as different parameters are unequal and some parameters are more critical for the performance on the pretraining task than others. The importance of a parameter can be represented by posterior distribution $p(\theta|D_p)$ near $\theta_0$ on the pretraining data $D_p$ that corresponds to the pretraining loss $\mathcal{L}(D_p;\theta) \propto p(D_p|\theta)$. In this way, elastic weight consolidation (EWC) (Kirkpatrick et al., 2017) assigns a Gaussian prior $\mathcal{N}(\theta_0, \sigma^2 I)$ with diagonal covariance $\sigma_i$ according to the estimated posterior distribution (i.e. loss landscape) of $\theta_i$ on the pretraining task. A parameter $\theta_i$ with larger impact to the $\mathcal{L}(D_p;\theta)$ will have sharper $p(\theta_i|D_p)$ and smaller $\sigma_i^2 = 1/(\alpha F_i)$, thus less flexibility in fine-tuning, lower variance in the fine-tuning prior, and higher weight for its L2 regularizer under the goal of preserving knowledge for the pretraining task.

To estimate this posterior distribution or loss landscape on the pretraining data $D_p$, we can perform Taylor expansion for the log likelihood $\log f(\theta) = \log p(\theta|D_p)$ near the parameters after pretraining, namely $\theta_0$, which is assumed to be near to the optimum, making $\nabla \log f(\theta_0) \approx 0$. Hence,

$$
\begin{aligned}
\log f(\theta) &= \log f(\theta_0) + \nabla \log f(\theta_0)(\theta - \theta_0) \\
&\quad + \frac{1}{2}(\theta - \theta_0)^T H_{\log f}(\theta_0)(\theta - \theta_0) + \cdots \\
&\approx \log f(\theta_0) + \frac{1}{2}(\theta - \theta_0)^T H_{\log f}(\theta_0)(\theta - \theta_0)
\end{aligned}
\tag{3}
$$

Therefore, through a second-order expansion, $p(\theta|D_p)$ is approximated by a Gaussian distribution corresponding to the negation of the quadratic term above, with $\theta_0$ being the mean and the Hessian matrix corresponding to the inverse covariance. To estimate the Hessian matrix, we use Bayes' theorem and take a flat prior on $D_p$, forming

$$
\begin{aligned}
H_{\log f}(\theta) &= \frac{\partial^2 \log p(\theta|D_p)}{\partial \theta^2} \\
&= \frac{\partial^2 \log p(D_p|\theta)}{\partial \theta^2} \\
&= \mathrm{E}_{x \sim p(x|\theta)}\big[\frac{\partial^2 \log p(x|\theta)}{\partial \theta^2}\big]
\end{aligned}
\tag{4}
$$

While the Fisher information matrix can be written as

$$
F = -\mathrm{E}_{x \sim p(x|\theta)}\big[\frac{\partial^2 \log p(x|\theta)}{\partial \theta^2}\big],
\tag{5}
$$

Therefore, the posterior distribution of the parameter $\theta$ on the pretraining task is approximated by a Gaussian distribution with the mean $\mu = \theta_0$ and the inverse covariance $\Sigma^{-1} = F$. The Fisher matrix can then be estimated by squared gradients as in Pascanu and Bengio (2014), and EWC further simplifies it by only considering diagonal terms.

## B  Implementation Details

We pretrain the model following the common settings in the field on single 24GB V100 GPUs using the Adam optimizer with a learning rate schedule of 20k linear warmup steps from 0 to 1e-4, followed by an inverse-sqrt decay to 3e-5. Models are selected and early stopping is performed according to the WER or BLEU on the *dev* set. 5-beam search is used during evaluation. The PTLMs we use are the 24-layer "large" versions provided by Hugging Face. A dynamic batching strategy is adopted to accommodate input utterance with different lengths. Accompanied with gradient accumulation, an average batch size of $\sim$25 with $\sim$500 target tokens

| SLURP Script | Label | ASR | ST | ST+ASR |
|---|---|---|---|---|
| is there a meeting on my calendar this afternoon | calendar,query | calendar,set | calendar,query | calendar,query |
| look for apple pie recipe | cooking,recipe | qa,stock | qa,recipe | cooking,recipe |
| can you really see russia from alaska | qa,factoid | alarm,remove | qa,factoid | general,quirky |
| are there morning shows available | calendar,query | weather,query | recommendation, events | lists,query |

Table 8: Examples of SLURP IC benchmark and predictions produced by different models.

per step is used. The wav2vec2 part is frozen for the first 10k steps, and utterances shorter than 0.1s or longer than 10s are not used during the first 20k steps. The L2 regularization with $\alpha$=5e-3 is applied to the weights, except the Bayesian transfer learning experiments. The setting is similar for the fine-tuning cases except that the encoder is frozen during the initial steps, and for joint-training models a 1:3 ratio between data for the pretraining and target task is used. While for smaller datasets including MINDS-14, SLURP-Fr, and Spoken Gigawords, the data ratio, dropout rate, and learning rate schedule are further tuned to avoid overfitting. We also build and compare with several cascaded pipelines based on our ST model, for which we directly use the model outputs with beam search without external LM as the model already leverages a strong language model. More details could be found from the source code.

## C Examples

Several examples in the SLURP IC benchmark and the predictions from different models are provided here in Table 8 for a more direct demonstration for the understanding capability of the models.

## D Dataset Details

Three new datasets are introduced in this work. Among them, SLURP-Fr is our main dataset for experiments on cross-lingual transfer, while we additionally carry out a series of experiments on Spanish (Es) to show that our methods work on more than one language. For the synthetic portion of SLURP-Fr/Es, we built the dataset based on MASSIVE, textual translation of SLURP, each using 4 speakers from Google TTS, with total 11.3H and 13.9H audio respectively. Therefore the contents are identical to MASSIVE. While for the real portion of SLURP-Fr, we leverage two native French

|  | train | dev | test | real |
|---|---|---|---|---|
| #Samples | 11514 | 2033 | 2974 | 477 |
| Avg. sec (Fr) | 2.47 | 2.44 | 2.44 | 2.35 |
| Avg. sec (Es) | 3.03 | 3.01 | 3.01 | / |

Table 9: Statistics of the SLURP-Fr/Es datasets.

|  | train | dev | test |
|---|---|---|---|
| #Samples | 50000 | 1000 | 385 |
| Mean length (sec) | 9.21 | 9.30 | 9.24 |
| Article word count | 23.9 | 24.0 | 24.0 |
| Headline word count | 7.8 | 8.1 | 8.0 |

Table 10: Statistics of the Spoken Gigaword dataset.

speakers to read the held-out samples from MASSIVE. The dataset size and mean length (in seconds) of the utterances are given in Table 9.

As for speech summarization, we follow MTL-SLT (Huang et al., 2022) to build the synthetic spoken version of Gigaword (Rush et al., 2015), using 9 speakers from Google TTS, with total 131.5H audio. We follow the data split in the original Gigaword dataset with a small and noisy test split (which we further filtered) and randomly sample from the train and dev split. The resultant size and mean length of the utterances are given in Table 10.

## E Spanish Experiments

Similar to the default type of models using only data with English and French, we first introduce the Spanish data to the En+Fr ASR model or the En↔Fr ST model to pretrain a En+Fr+ES ASR model as well as a En↔Fr+En↔Es ST model, with satisfactory results as in Table 11. Both ASR and ST models are then fine-tuned with joint training on SLURP, reaching 87.63% and 89.59% accuracy respectively. Then they are used for cross-lingual

|            | TEDx    | MuST-C  | CoVoST2 |
|------------|---------|---------|---------|
| ASR WER↓   | 15.05%  | 8.94%   | 11.34%  |
| ST BLEU↑   | 25.84%  | 30.06%  | 33.90%  |

Table 11: Test results of the model with Spanish data added on the cleaned pretraining datasets given by word error rate (WER) for ASR and BLEU score for ST, with Spanish inputs for TEDx and CoVoST2, and English inputs for MuST-C.

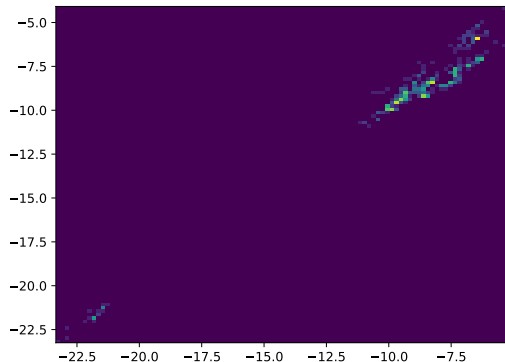

Figure 3: The distribution of the estimated Fisher diagonals shown in heat map, with x-axis for the means of the log squared gradients of each weight or bias, and y-axis the standard deviation.

transfer to SLURP-Es.

## F  EWC Weight Distribution

The distributions of the log estimated Fisher diagonals for each weight matrix or bias vector are illustrated in Figure 3. It can be observed that most weights are concentrated around 1e-5 to 1e-10, and they are close to each other as the standard deviations are at the similar magnitude. Hence with $\alpha$=1e-7, most weights will reach the 1e-2 clamping threshold. The exceptions are the biases for key projection in attention modules, which correspond to the lower-left cluster and have much smaller weights.