# OpenReview forum: "The Interpreter Understands Your Meaning: End-to-end Spoken Language Understanding Aided by Speech Translation"
_EMNLP/2023/Conference — EMNLP 2023 Findings_

### Official Review · Reviewer_1hZ9 · 2023-08-04

**Paper Topic And Main Contributions:** 1. The paper demonstrates the effecti…
**Typos Grammar Style And Presentation Improvements:** In line 227, "with ∼600 utterances"
**Soundness:** 3

**Excitement:**

3: Ambivalent: It has merits (e.g., it reports state-of-the-art results, the idea is nice), but there are key weaknesses (e.g., it describes incremental work), and it can significantly benefit from another round of revision. However, I won't object to accepting it if my co-reviewers champion it.

**Missing References:**

[1] Chang Y H, Chen Y N. Contrastive Learning for Improving ASR Robustness in Spoken Language Understanding[J]. arXiv preprint arXiv:2205.00693, 2022.
[2] Hsu C J, Chung H L, Lee H, et al. T5lephone: Bridging Speech and Text Self-supervised Models for Spoken Language Understanding via Phoneme level T5[J]. arXiv preprint arXiv:2211.00586, 2022.

**Questions For The Authors:**

1. Could you explain why whisper is not suitable for fine-tuning on SLURP?
2. Could you provide the statistics of your synthetic datasets?

**Reasons To Accept:**

1. The paper provides the experimental results on a large number of downstream tasks.
2. The method achieves consistent performance improvement on these downstream tasks.
3. The paper builds some new datasets, which may be helpful to the community.

**Reasons To Reject:**

1. For each task, the paper only provides the experimental results of one dataset. For example, the paper could provide the results on ATIS and TREC6 for English SLU as in [1].
2. The paper would be better if it provided some cases to show the effectiveness of your method.
3. It is a little unprofessional to put all the experimental results of speech summarization in the appendix.

[1] Chang Y H, Chen Y N. Contrastive Learning for Improving ASR Robustness in Spoken Language Understanding[J]. arXiv preprint arXiv:2205.00693, 2022.

**Reproducibility:**

3: Could reproduce the results with some difficulty. The settings of parameters are underspecified or subjectively determined; the training/evaluation data are not widely available.

**Reviewer Confidence:**

4: Quite sure. I tried to check the important points carefully. It's unlikely, though conceivable, that I missed something that should affect my ratings.

---

> ### Author Rebuttal · Authors · 2023-08-29
>
> Thank you very much for your review. We have tried to make the manuscript more consistent and hypothesis based, which addresses some concerns. Here we also address some criticisms directly:
>
> * Reasons To Reject:
>
> >1. For each task, the paper only provides the experimental results of one dataset. For example, the paper could provide the results on ATIS and TREC6 for English SLU as in [1].
>
> We choose SLURP as the benchmark for evaluation on the English intent classification (IC) task, since SLURP is the only recently-proposed benchmark on the standard IC task that remains challenging, while the state-of-the-art in other previous well-known datasets like ATIS, SNIPS, and FSC has long reached 99% accuracy, making the empirical comparison largely random. To simplify the problem and to emphasize the evaluation of the pretrained encoder's semantic understanding capability, we largely focus on the classification task without the need of building a task-specific decoder.  Since we work on end-to-end SLU from audio, benchmarks like TREC6 with only textual inputs are not suitable.
>
> It is also noteworthy that we perform experiments on English IC (SLURP), multi-lingual IC (MINDS-14) and cross-lingual IC (SLURP-Fr/Es), and MINDS-14 contains English portion. Therefore, as for the IC task, various situations and aspects have been investigated using multiple datasets.
>
> >2. The paper would be better if it provided some cases to show the effectiveness of your method.
>
> The effectiveness of our methods has been largely demonstrated by the accuracy number. While several typical examples from SLURP are shown below and added to the appendix:
>
> | SLURP Script                                     | Label          | ASR           |          ST           |     ST+ASR     |
> | ------------------------------------------------ | -------------- | ------------- | :-------------------: | :------------: |
> | is there a meeting on my calendar this afternoon | calendar,query | calendar,set  |    calendar,query     | calendar,query |
> | look for apple pie recipe                        | cooking,recipe | qa,stock      |       qa,recipe       | cooking,recipe |
> | can you really see russia from alaska            | qa,factoid     | alarm,remove  |      qa,factoid       | general,quirky |
> | are there morning shows available                | calendar,query | weather,query | recommendation,events |  lists,query   |
>
>
>
> Below are some examples from the Spoken Gigaword dataset:
>
> > Article: arsenal manager arsene wenger was charged by the football association (fa) on monday with improper conduct over a reported comment about manchester united striker ruud van nistelrooy.
> >
> > Label: wenger charged with improper conduct
> >
> > ASR: wenger charged over van nistelrooy comments
> >
> > ST: wenger charged with misconduct over manchester united comment
> >
> > ST+ASR: arsenal manager winger charged with improper conduct
> >
> > Article: india's top nuclear expert shrugged off antinuclear protests by indians on hiroshima day, saying the activists should instead shout slogans in washington and moscow, a newspaper reported saturday.
> >
> > Label: top nuclear scientist shrugs off indian antinuclear protests
> >
> > ASR: india says activists should instead shoot slogans in washington moscow
> >
> > ST: india's top nuclear expert calls for anti-nuclear protests
> >
> > ST+ASR: india's top nuclear expert warns against anti-nuclear protests
> >
> > Article: syria and lebanon decided on monday to boost border controls and anti-terrorism coordination, as the two neighbors took a new step to strengthen ties since diplomatic relations were established.
> >
> > Label: syria and lebanon to boost border anti-terror controls
> >
> > ASR: syria lebanon boost border controls
> >
> > ST: syria lebanon to boost border controls
> >
> > ST+ASR: syria lebanon to boost border controls and anti-terrorism coordination
> >
> > Article: austrian women in leading positions complained about lingering male domination in their society in a meeting tuesday with visiting us first lady hillary rodham clinton.
> >
> > Label: austrian women complain to mrs. clinton about male domination by roland prinz
> >
> > ASR: austrian women leaders meet clinton
> >
> > ST: austrian women complain about lingering male domination
> >
> > ST+ASR: austria women complain about lingering male-domination in society
>
>
>
> >3. It is a little unprofessional to put all the experimental results of speech summarization in the appendix.
>
> This was also raised by other reviewers; we have updated the manuscript to move details on self-supervised learning on NLP and pretrained knowledge preservation to the related work section and the appendix, while the results on speech summarization to the main article.
>
> * Questions For The Authors:
>
> >1. Could you explain why whisper is not suitable for fine-tuning on SLURP?
>
> It is very difficult to determine the reason why a baseline model is not working well, but as for Whisper there are several possible reasons.
>
> 1. Whisper is trained on En ASR and X->En ST but not for En->X ST. Our hypothesis is that the ST pretraining on a specific language would enhance the semantic understanding capabilities of the model in that language, which may not help Whisper much on the English SLURP benchmark.
> 2. Whisper is trained on 30s chunks. When used to transcribe relatively short segments, Whisper often produces transcription with significant error, or simply hallucinates. While the SLU datasets we are using contain mostly short (<10s) utterances, hence this domain shift may affect Whisper's performance. The situation could be partially alleviated by padding all of them to 30s, but it would lead to a significant waste of computational power that we are unable to afford.
>
> We nevertheless report the results of the Whisper-based model as it is an important concurrent pretraining work.
>
>
>
> >2. Could you provide the statistics of your synthetic datasets?
>
> Thank you for pointing out this missing part in our manuscript, and we have added the explanation and statistics of our new datasets below.
>
> SLURP-Fr is our main dataset for experiments on cross-lingual transfer, while we additionally carry out a series of experiments on Spanish (Es) to show that our methods work on more than one language.  For the synthetic portion of SLURP-Fr/Es, we built the dataset based on MASSIVE, textual translation of SLURP, each using 4 speakers from Google TTS, with total 11.3H and 13.9H audio respectively. Therefore the contents are identical to MASSIVE. While for the real portion of SLURP-Fr, we leverage two native French speakers to read the held-out samples from MASSIVE. The dataset size and mean length of the utterances are given below:
>
>
> |                        | train |  dev | test | real |
> |------------------------|-------|:----:|:----:|:----:|
> | #Samples               | 11514 | 2033 | 2974 |  477 |
> | Mean length (sec) - Fr | 2.47  | 2.44 | 2.44 | 2.35 |
> | Mean length (sec) - Es | 3.03  | 3.01 | 3.01 |   /  |
>
>
>
>
> As for speech summarization, we follow [MTL-SLT](https://aclanthology.org/2022.nlp4convai-1.11/) to build the synthetic spoken version of [Gigaword](https://huggingface.co/datasets/gigaword), using 9 speakers from Google TTS, with total 131.5H audio. We follow the data split in the original Gigaword dataset with a small and noisy test split (which we further filtered) and randomly sample from the train and dev split. The resultant size and mean length of the utterances are given below:
>
>
> |                        | train |  dev | test |
> |------------------------|-------|:----:|:----:|
> | #Samples               | 50000 | 1000 |  385 |
> | Mean length (sec) | 9.21  | 9.30 | 9.24 |
> | Article word count      | 23.9  | 24.0 | 24.0 |
> | Headline word count      | 7.8   | 8.1  | 8.0  |
>
>
>
>
> * Missing References:
>
> >[1] Chang Y H, Chen Y N. Contrastive Learning for Improving ASR Robustness in Spoken Language Understanding[J]. arXiv preprint arXiv:2205.00693, 2022. [2] Hsu C J, Chung H L, Lee H, et al. T5lephone: Bridging Speech and Text Self-supervised Models for Spoken Language Understanding via Phoneme level T5[J]. arXiv preprint arXiv:2211.00586, 2022.
>
> Thank you for the suggestions and we have added the references accordingly.
>
>
>
> * Typos Grammar Style And Presentation Improvements:
>
> > In line 227, "with ∼600 utterances"
>
> We have fixed them in the manuscript.

---

### Official Review · Reviewer_bU2d · 2023-08-05

**Soundness:** 3

**Excitement:**

3: Ambivalent: It has merits (e.g., it reports state-of-the-art results, the idea is nice), but there are key weaknesses (e.g., it describes incremental work), and it can significantly benefit from another round of revision. However, I won't object to accepting it if my co-reviewers champion it.

**Paper Topic And Main Contributions:**

The paper proposes to use speech translation for pretraining E2E spoken language understanding models for both intra and cross-lingual scenarios.

The approach improved the performance on a series tasks and benchmarks, and the paper also proposes new benchmark datasets for 	speech summarization and low/zero-resource cross-lingual transfer.

**Reasons To Accept:**

The approach using ST for pretraining SLU tasks is well-motivated and the motivations are clearly presented.

The work tested the pretrained model with several downstream tasks, the preservation of pretrained knowledge and created synthetic datasets for two tasks.

The authors also stated that the code and models will be open-sourced.

**Reasons To Reject:**

Some experimental settings, analysis and comparisons are not always clearly explained, and could benefit from some reorganization. For instance, in Table 4, the DUAL approach and the approach in the paper are using different types spans (document/segment), thus seem hardly directly comparable. Also, the caption claims that ST pretraining is outperforming the cascaded system, but it is not always the case (on dev).

The writing and presentation are not always easy to follow; the work may benefit from some better organization of the presentation flow. For instance, the investigation of speech summarization seems quite relevant to the main contributions described in the paper.

**Reproducibility:**

4: Could mostly reproduce the results, but there may be some variation because of sample variance or minor variations in their interpretation of the protocol or method.

**Reviewer Confidence:**

3: Pretty sure, but there's a chance I missed something. Although I have a good feel for this area in general, I did not carefully check the paper's details, e.g., the math, experimental design, or novelty.

**Typos Grammar Style And Presentation Improvements:**

Grammar:
- l.183: "we've also attempt" -> "attempted"

Style:
- l.172, the sentence starting with 'While...' is not clearly connected to the sentence it's supposed to contrast with.

---

> ### Author Rebuttal · Authors · 2023-08-29
>
> Thank you very much for your review. We have tried to make the manuscript more consistent and hypothesis based, which addresses some concerns.  Here we also address some criticisms directly:
>
> * Reasons To Reject:
>
> > Some experimental settings, analysis and comparisons are not always clearly explained, and could benefit from some reorganization. For instance, in Table 4, the DUAL approach and the approach in the paper are using different types spans (document/segment), thus seem hardly directly comparable. Also, the caption claims that ST pretraining is outperforming the cascaded system, but it is not always the case (on dev).
>
> Thank you for your comments and we have tried to provide more discussions in the manuscript. To summarize, our core hypothesis is that ST pretraining could introduce stronger semantic understanding capabilities to the model compared to ASR and self-supervised pretraining. Therefore all our experiments compare these 3 types of pretraining across different tasks, which show consistent results supporting our hypothesis.
>
>
>
> Specifically, the baseline DUAL model enjoys an advantage of the view on the whole document, while the scope of our model is limited to each segment in order to have an end-to-end architecture consistent with other experiments, different from DUAL using pre-extracted features from HuBERT on audio chunks. Hence we have a strong prior that the comparison is actually unfair to our model, and the surprisingly better results of ours further demonstrate the strong advantage of our model. We use the AOS results reported in the original DUAL paper with the document-level view. Unfortunately we are unable to fully reproduce DUAL at the moment using their code, or to further adapt the DUAL model to segment-level span and to report the performance of DUAL with segment-level view only.
>
>
>
> We acknowledge that ST pretraining has a close but not better results compared to the cascaded model (58.2% vs 58.3%), and we have updated the manuscript to clarify that. However it is noteworthy that in many previous papers only _test_ results on NMSQA are reported, since the test split is more challenging with real audios from multiple speakers. Therefore our better results on the test split are more meaningful.
>
>
> > The writing and presentation are not always easy to follow; the work may benefit from some better organization of the presentation flow. For instance, the investigation of speech summarization seems quite relevant to the main contributions described in the paper.
>
> This was also raised by other reviewers, and clearly makes more sense; we have updated the manuscript to move details on self-supervised learning on NLP and pretrained knowledge preservation to the related work and appendix, while the results on speech summarization to the main article. Also we further emphasize our hypotheses and the empirical evidence we use to support them.
>
> * Typos Grammar Style And Presentation Improvements:
>
> >Grammar:
> >l.183: "we've also attempt" -> "attempted"
> Style:
> >l.172, the sentence starting with 'While...' is not clearly connected to the sentence it's supposed to contrast with.
>
> We have fixed them in the manuscript.

---

### Official Review · Reviewer_LTf6 · 2023-08-05

**Typos Grammar Style And Presentation Improvements:** 1. Instead of None, maybe trained fro…
**Soundness:** 3

**Excitement:**

4: Strong: This paper deepens the understanding of some phenomenon or lowers the barriers to an existing research direction.

**Paper Topic And Main Contributions:**

The paper introduces a novel approach to pretraining speech models for end-to-end spoken language understanding (SLU) tasks using speech translation (ST). The authors argue that ST enables the models to capture high-level semantic information from input utterances and establish connections between different languages, which is not adequately achieved by pretrained spoken language models that focus on lower-level audio features. ST exhibits the ability to capture long-term dependencies instead of just local context information, making it more suitable for SLU tasks compared to automatic speech recognition (ASR). Additionally, ST demonstrates superior cross-lingual transfer capabilities, even outperforming multilingual ASR in zero-shot cases.

To validate their hypothesis, the authors conduct experiments on five different datasets, including two newly proposed datasets. The results demonstrate that introducing ST as a pretraining method leads to improved performance compared to baseline approaches and ASR-pretrained models. Though the results vary across datasets, the authors showcase the value of preserving ST-pretrained knowledge through Bayesian regularization, particularly under low-resource conditions.

In summary, the paper presents a promising approach to enhancing SLU performance by leveraging ST as a pretraining technique for speech models. The experiments on various datasets provide strong evidence in favor of the proposed method's effectiveness. The findings offer valuable insights into advancing SLU tasks.

**Questions For The Authors:**

1. What is Es stand for in Table2?

**Reasons To Accept:**

1. Sound assumption:
The paper builds upon a sound assumption by proposing to utilize speech translation (ST) as a pretraining method for speech models in end-to-end spoken language understanding (SLU) tasks. The authors' assumption, supported by evidence, suggests that ST can effectively capture high-level semantic information and cross-lingual associations, making it a promising approach to enhance SLU performance.

2. Strong Empirical Validation:
The paper presents thorough experimental validation of the proposed ST-pretraining approach on multiple datasets, including new benchmarks. The reported results consistently show that ST-pretrained models outperform ASR and acoustic pretrained models, indicating the effectiveness of the proposed approach.

3. Insights into Cross-Lingual Transfer:
The paper provides valuable insights into the cross-lingual transfer capabilities of the ST-pretrained models, particularly in zero-shot cases. This feature is of great practical importance, showcasing the potential for the proposed method to handle multilingual tasks effectively, even with limited training data in new languages.

**Reasons To Reject:**

1. Limited Generalizability: The efficacy of the proposed ST pretraining method could be limited to specific language pairs.

2. Lacking Quantitative  Discussion: While the paper presents quantitative results and comparisons, it lacks a comprehensive qualitative discussion on the specific strengths and weaknesses of the proposed ST-pretraining approach. For example, compared to ST, ST+ASR performance degrade across all the dataset, but the authors fail to discuss it.

3. Lacking Qualitative Analysis: While the paper demonstrates the superiority of ST-pretrained models over baseline methods, it falls short in providing a detailed quantitative comparison with ASR pretrained and acoustic pretrained models. A more comprehensive analysis directly comparing the performance of ST-pretrained models against ASR pretrained models and acoustic pretrained models would strengthen the paper's claims. For example, providing concrete examples where the ST-pretrained models succeed while other ASR pretrained models fail would be beneficial in highlighting the specific advantages of the proposed approach and addressing potential concerns about the lack of quantitative comparison.

4. The experiments are not unified across datasets: The paper's lack of a consistent experimental setup across different datasets may introduce confounding factors that could affect the validity of the results. Without standardized experiments, it becomes difficult to draw definitive conclusions and assess the true impact of the ST-pretraining approach on SLU performance. For example, only joint training +Es is applied on SLURP dataset. No joint training applied on Spoken QA.

5. Redundant context in the paper: The paper may suffer from redundant context in certain sections. This redundancy can obscure the main contributions or key takeaways. For example, the first paragraph in the introduction introduce the self-supervised learning method for NLP pretrained model, but it's irrelevant to the paper. Also, in section 5, the authors spend lots of space introducing L2-SP and EWC algorithm for pretrained knowledge preservation, but put the results of speech summarization to appendix. They should put the details of L2-SP and EWC to appendix but summarization results in main article instead. Additionally, essential methodologies like Adversarial training in the cross-lingual transfer setting should be elaborated upon in the main article







**Reproducibility:**

4: Could mostly reproduce the results, but there may be some variation because of sample variance or minor variations in their interpretation of the protocol or method.

**Reviewer Confidence:**

4: Quite sure. I tried to check the important points carefully. It's unlikely, though conceivable, that I missed something that should affect my ratings.

---

> ### Author Rebuttal · Authors · 2023-08-29
>
> Thank you very much for your review. We have tried to make the manuscript more consistent and hypothesis based, which addresses some concerns.  Here we also address some criticisms directly:
>
> * Reasons To Reject:
>
> >1. Limited Generalizability: The efficacy of the proposed ST pretraining method could be limited to specific language pairs.
>
> Admittedly,  ST data is only available in some language pairs, as mentioned in L108~114 in the manuscript. However, for each covered language, there are a large number of diverse downstream SLU tasks with only rich data in English. Therefore, it is a practical need to enroll various such languages to an English-only system trained on each specific SLU task. This goal of cross-lingual transfer of specific English SLU models to other languages is one of the main motivation of our work. Furthermore, following our hypothesis, ST pretraining could also enhance the results on the source language (En), which is validated by our experiments.
>
> It is also noteworthy that, in order to show that our method does not only work on French as the target language, we had an additional series of experiments using Spanish (_Es_) as the target: ST/ASR pretraining, training on SLURP (En), and then transfer to the target language with SLURP-Es. Positive results with better accuracy on SLURP (Table 2) and similar strong cross-lingual performance (Table 6) indicate that our methods could be applied to different target languages.
>
>
> >2. Lacking Quantitative Discussion: While the paper presents quantitative results and comparisons, it lacks a comprehensive qualitative discussion on the specific strengths and weaknesses of the proposed ST-pretraining approach. For example, compared to ST, ST+ASR performance degrade across all the dataset, but the authors fail to discuss it.
>
> We have added discussion to the manuscript to address this concern. In summary, our hypothesis is that ST pretraining endows the model stronger semantic understanding capabilities compared to ASR or acoustic pretraining, while additional multi-tasking with the less semantic-demanding ASR may have possibly negative but limited impact, unless it is favorable for some specific downstream task. This matches the experimental results: ST+ASR leads to performance close to ST in most cases (slightly worse in SLURP but slightly better w/ Joint training, worse than ST in MINDS-14 but the gap is small w/ Joint training, slightly worse in NMSQA, varied in SLURP-Fr), but consistently better than ASR-only models. While for the specific task of speech summarization which is more similar to ASR to produce textual summarization in English, ST+ASR could get the best results.
>
>
>
> >3. Lacking Qualitative Analysis: While the paper demonstrates the superiority of ST-pretrained models over baseline methods, it falls short in providing a detailed quantitative comparison with ASR pretrained and acoustic pretrained models. A more comprehensive analysis directly comparing the performance of ST-pretrained models against ASR pretrained models and acoustic pretrained models would strengthen the paper's claims. For example, providing concrete examples where the ST-pretrained models succeed while other ASR pretrained models fail would be beneficial in highlighting the specific advantages of the proposed approach and addressing potential concerns about the lack of quantitative comparison.
>
> Our core hypothesis is that ST pretraining will lead to better SLU performance in intra-lingual, cross-lingual, and multi-lingual situations, compared to baseline models using ASR pretraining and only self-supervised/acoustic-textual pretraining. Our ST-pretraining approach is directly compared with ASR and self-supervised pretrained models in all our experiments. Similar clarification and emphasis of the results and their implications to our hypothesis have been added to the manuscript.
>
> All the evaluated benchmarks provide examples of the scenarios where the proposed ST pretraining approach outperforms ASR and self-supervised pretraining, as reflected by the accuracy numbers. While several typical examples from SLURP are shown below and added to the appendix:
>
>
> | SLURP Script                                           | Label          | ASR          |       ST       |     ST+ASR     |
> |--------------------------------------------------|----------------|---------------|:---------------------:|:--------------:|
> | is there a meeting on my calendar this afternoon | calendar,query | calendar,set  |     calendar,query    | calendar,query |
> | look for apple pie recipe                        | cooking,recipe | qa,stock      |       qa,recipe       | cooking,recipe |
> | can you really see russia from alaska            | qa,factoid     | alarm,remove  |       qa,factoid      | general,quirky |
> | are there morning shows available                | calendar,query | weather,query | recommendation,events | lists,query    |
>
>
>
> Below are some examples from the Spoken Gigaword dataset:
>
> > Article: arsenal manager arsene wenger was charged by the football association (fa) on monday with improper conduct over a reported comment about manchester united striker ruud van nistelrooy.
> >
> > Label: wenger charged with improper conduct
> >
> > ASR: wenger charged over van nistelrooy comments
> >
> > ST: wenger charged with misconduct over manchester united comment
> >
> > ST+ASR: arsenal manager winger charged with improper conduct
> >
> > Article: india's top nuclear expert shrugged off antinuclear protests by indians on hiroshima day, saying the activists should instead shout slogans in washington and moscow, a newspaper reported saturday.
> >
> > Label: top nuclear scientist shrugs off indian antinuclear protests
> >
> > ASR: india says activists should instead shoot slogans in washington moscow
> >
> > ST: india's top nuclear expert calls for anti-nuclear protests
> >
> > ST+ASR: india's top nuclear expert warns against anti-nuclear protests
> >
> > Article: syria and lebanon decided on monday to boost border controls and anti-terrorism coordination, as the two neighbors took a new step to strengthen ties since diplomatic relations were established.
> >
> > Label: syria and lebanon to boost border anti-terror controls
> >
> > ASR: syria lebanon boost border controls
> >
> > ST: syria lebanon to boost border controls
> >
> > ST+ASR: syria lebanon to boost border controls and anti-terrorism coordination
> >
> > Article: austrian women in leading positions complained about lingering male domination in their society in a meeting tuesday with visiting us first lady hillary rodham clinton.
> >
> > Label: austrian women complain to mrs. clinton about male domination by roland prinz
> >
> > ASR: austrian women leaders meet clinton
> >
> > ST: austrian women complain about lingering male domination
> >
> > ST+ASR: austria women complain about lingering male-domination in society
>
>
>
>
> >4. The experiments are not unified across datasets: The paper's lack of a consistent experimental setup across different datasets may introduce confounding factors that could affect the validity of the results. Without standardized experiments, it becomes difficult to draw definitive conclusions and assess the true impact of the ST-pretraining approach on SLU performance. For example, only joint training +Es is applied on SLURP dataset. No joint training applied on Spoken QA.
>
> Thank you for pointing out that, and we try to further clarify the point below and in the manuscript: As for the response to point 3 above, the core experiments (ST vs ASR/self-supervised pretraining) are consistent across the datasets. While there are several additional experiments to support our hypothesis in other aspects. For example, an additional series of +Es experiments is used to show that our method works not only in the French language as mentioned in L128. While as mentioned in L361~362, most of the encoder layers are frozen in the Spoken QA experiment to make a fair comparison with the baseline (DUAL) regarding the model size while maintain a model architecture consistent with experiments on other tasks. Therefore, joint training is less meaningful in this situation since most parameters shared between the source task and the target QA task are frozen and will not be trained.
>
>
> >5. Redundant context in the paper: The paper may suffer from redundant context in certain sections. This redundancy can obscure the main contributions or key takeaways. For example, the first paragraph in the introduction introduce the self-supervised learning method for NLP pretrained model, but it's irrelevant to the paper. Also, in section 5, the authors spend lots of space introducing L2-SP and EWC algorithm for pretrained knowledge preservation, but put the results of speech summarization to appendix. They should put the details of L2-SP and EWC to appendix but summarization results in main article instead. Additionally, essential methodologies like Adversarial training in the cross-lingual transfer setting should be elaborated upon in the main article
>
> This was also raised by other reviewers, and clearly makes more sense; we have updated the manuscript to move details on self-supervised learning on NLP and pretrained knowledge preservation to the related work and appendix, while the results on speech summarization and explanations on adversarial training to the main article. While it is noteworthy that adversarial training is an extra trick to enhance our technique and does not affect our conclusion.
>
>
> * Questions For The Authors:
>
> >1. What is Es stand for in Table2?
>
> Es stands for Spanish. Additional clarification is added to the manuscript.
>
> * Typos Grammar Style And Presentation Improvements:
> >1. Instead of None, maybe trained from scratch is better term in each table.
>
> Thank you for the suggestion and we have updated the manuscript accordingly. It should be noted that the _None_ models use acoustic and textual pretraining based on XLSR-53 and mBART, instead of completely training from scratch on the target task.

---

### Meta-Review · Area_Chair_Shod · 2023-09-23

**Recommendation:** 4

**Metareview:**

The paper presents a new approach to using speech translation (where data is relatively abundant) for pre-training E2E spoken language understanding models for both intra and cross-lingual scenarios. While not a revolutionary novel idea, the approach improves performance on a series of existing tasks and benchmarks (setting SOTA performance levels on some), and the paper also proposes new benchmark datasets for speech summarization and low/zero-resource cross-lingual transfer. Authors also commit to releasing the code and models as open-sourced, which will create an interesting and relevant test-bed for future development. Authors responded well to reviewers (whose main criticism is the organization of the paper, and the presentation of results, sometimes asking for more detail, which the authors generally provide in the response), and commit to incorporating feedback and improve readability, which I think may even allow including the approach in the main conference, certainly in Findings.

---

### Decision · Program_Chairs · 2023-10-07

**Decision:**

Accept-Findings

**Comment:**

The paper presents a new approach to using speech translation (where data is relatively abundant) for pre-training E2E spoken language understanding models for both intra and cross-lingual scenarios. While not a revolutionary novel idea, the approach improves performance on a series of existing tasks and benchmarks (setting SOTA performance levels on some), and the paper also proposes new benchmark datasets for speech summarization and low/zero-resource cross-lingual transfer. Authors also commit to releasing the code and models as open-sourced, which will create an interesting and relevant test-bed for future development. Authors responded well to reviewers (whose main criticism is the organization of the paper, and the presentation of results, sometimes asking for more detail, which the authors generally provide in the response), and commit to incorporating feedback and improve readability, which I think may even allow including the approach in the main conference, certainly in Findings.